# Investigation of the Potential Effects of Host Genetics and Probiotic Treatment on the Gut Bacterial Community Composition of Aquaculture-raised Pacific Whiteleg Shrimp, *Litopenaeus vannamei*

**DOI:** 10.3390/microorganisms7080217

**Published:** 2019-07-26

**Authors:** Angela Landsman, Benoit St-Pierre, Misael Rosales-Leija, Michael Brown, William Gibbons

**Affiliations:** 1trū Shrimp Innovation Center, The trū Shrimp Company, 330 3rd Street, Balaton, MN 56115, USA; 2Department of Biology and Microbiology, South Dakota State University, Alfred Dairy Science Hall, Box 2104A, 1224 Medary Avenue, Brookings, SD 57007, USA; 3Department of Animal Science, South Dakota State University, Animal Science Complex, Box 2170, Brookings, SD 57007, USA; 4Department of Natural Resource Management, South Dakota State University, Edgar S. McFadden Biostress Lab, Box 2140B, 1390 College Avenue, Brookings, SD 57007, USA

**Keywords:** aquaculture, intestinal microbiome, Pacific whiteleg shrimp, genetic background, probiotic

## Abstract

This study presents the potential effects of the genetic background and use of probiotics on the gut bacterial composition of Pacific whiteleg shrimp (*Litopenaeus vannamei*) grown in an indoor aquaculture facility. The strains investigated were Shrimp Improvement Systems (SIS, Islamorada, FL, USA), a strain genetically selected for disease resistance, and an Oceanic Institute (OI, Oahu, HI, USA) strain, selected for growth performance. BioWish 3P (BioWish Technologies, Cincinnati, OH, USA) was the selected probiotic. The study consisted of two separate trials, where all shrimp were raised under standard industry conditions and fed the same diet. Shrimp were stocked in 2920 L production tanks at a density of 200/m^3^ and acclimated for 14 days. After the acclimation period, triplicate tanks were supplemented daily for a duration of 28 days with probiotics, while three other tanks did not receive any treatment (controls). During the 28-day trial period, there was no statistically supported difference (*p* > 0.05) in either performance or health status as a result of genetic background or probiotic treatment. However, differences in gut bacterial composition, as assessed by high throughput sequencing of amplicons generated from the V1-V3 region of the bacterial 16S rRNA gene, were observed. The relative abundance of five major operational taxonomic units (OTUs) were found to vary significantly across experimental groups (*p* < 0.05). Notably, operational taxonomic unit (OTU) SD_Shr-00006 was at its highest abundance in d43 SIS samples, with levels greater than d71 samples of the same genetic line or any of the OI shrimp samples. OTUs for SD_Shr-00098 displayed a similar type of profile, but with highest abundance in the OI genetic line and lowest in the SIS shrimp. SD_Shr-00004 showed an opposite profile, with highest abundance in the SIS d71 samples and lowest in the SIS d43 samples. Together, these results suggest that host genetic background can be an important determinant of gut bacterial composition in aquaculture-raised whiteleg shrimp and indicate that development of strategies to manipulate the microbiome of this important seafood will likely need to be customized depending on the genetic line.

## 1. Introduction

Shrimp represent the most valuable seafood in the world, with a hold on 78% of the seafood monetary market [1]. This industry has managed to grow despite stagnant yields in wild-caught harvests through a substantial increase in aquaculture production. Indeed, 55% percent of the annual global shrimp supply in 2018 was produced through commercial farming [2], indicating that aquaculture has the capacity to provide consumers with a consistent and reliable supply of product [3]. Shrimp farming has shown great potential for high productivity at reduced costs. Notably, aquaculture-raised shrimp have shown twice the growth rates of wild stocks, indicating great potential to further increase production [4]. However, periodic disease outbreaks and continued animal health management problems have become a great concern to potential investors, and consequently to the future of aquaculture. In order to mitigate losses in production due to disease, some major producers have opted to generate greater quantities of smaller-sized product by harvesting shrimp at an earlier stage, before significant losses can occur. However, flooding the markets with such lower value product can only be viable in the short term, as it may impact consumer demand [1]. As the market for shrimp continues to increase worldwide, the aquaculture industry needs to continue developing innovative strategies to take advantage of market growth opportunities.

Two main types of production systems are generally used in shrimp aquaculture. Outdoor systems, which typically consist of ponds that rely on natural saltwater sources for maintaining optimal growth conditions, offer a low-cost opportunity for shrimp aquaculture. In contrast, the more costly indoor facilities allow tighter biosecurity control, safer products and reduced environmental footprint as they use recirculating water systems with limited exposure to natural environments [5]. The most widely used species worldwide for both outdoor and indoor production systems is the whiteleg shrimp (*Litopenaeus vannamei*), also known as Pacific white shrimp or king prawn, as it expresses a number of desirable traits, such as tolerance to a wide range of salinities and temperatures [6,7]. Overall, the development of shrimp genetic lines has focused mainly on increased growth and higher disease resistance, as these two traits are of major importance for aquaculture production. However, because they are mutually exclusive in inheritance [8], breeding programs typically focus on one of these traits depending on the production system. Because of greater risks of exposure to opportunistic pathogens, disease resistance is of higher importance for outdoor systems, while genetic line development for bio-secure indoor systems has focused on increasing shrimp growth as exposure to environmental pathogens is greatly reduced [8]. The use of rapidly growing shrimp with reduced disease resistance comes at a risk, however, as systemic stressors can promote the growth of ubiquitously present *Pseudomonas* or *Vibrio* species that can colonize the shrimp gut and cause disease [9]. One possible strategy to mitigate risk of disease in susceptible genetic lines would be to promote health through microbial management [5]. Considering the overarching importance of the gut microbiome in animal health and nutrition [10], manipulating beneficial gut microbial communities could potentially improve aquaculture shrimp resistance to pathogens without prophylactic use of antibiotics. 

Research to date has found that the gut bacterial profile of healthy shrimp consists primarily of Proteobacteria [11]. This is consistent with reported profiles in marine fish, but in stark contrast to the microbiome of terrestrial animals in which Firmicutes and Bacteroides are typically dominant [12]. Indeed, these latter phyla have so far been found to be minor components of the shrimp gut microbiome, and their abundance appears to be highly dependent on local environmental conditions and diet composition [9]. While great strides have so far been made towards defining the composition of gut bacterial communities in healthy shrimp, there is a critical need to determine how composition is impacted by production parameters. Based on a wide body of research literature on human and terrestrial animal microbiomes, host genetics is likely an important factor affecting the composition of gut symbionts in shrimp. A number of management practices also have the potential to influence gut bacterial community profiles, but there is limited in-depth knowledge to date on the effect of aquaculture practices on the shrimp gut microbiome [10]. For instance, the biosafety measures implemented to prevent pathogen outbreaks, such as water and feed sterilization, may also inadvertently affect gut colonization of indoor raised shrimp as these procedures limit exposure to beneficial bacteria found in natural environments [10]. Absence of certain microorganisms during early events of microbiome development in shrimp raised indoor could possibly impact future performance, productivity or disease resistance [13]. Other factors that can affect gut bacterial community profiles include water chemistry and diet composition. For instance, feeds being offered in aquaculture are trending towards formulations with increased proportions of less expensive ingredients, such as plant proteins, fiber, and carbohydrates, which are not natural components of shrimp diets [13]. Finally, the use of probiotics can also possibly affect the composition of gut bacterial communities. Probiotics have historically been used for competitive exclusion of pathogens during stress events in pond systems [14], but they also represent a potential means of transitioning gut microbiomes for improved gut health and digestion of plant-based feed ingredients.

In light of the critical need to gain further insight on the influence of host genetics and management practices on the shrimp microbiome, the research presented in this report aimed to investigate the possible impact of host genetic and probiotic use on modulating the composition of the whiteleg shrimp microbiome. To this end, the gut bacterial profiles of whiteleg shrimp raised in an indoor facility with the same diet were determined from two different genetic lines, with or without supplementation with commercial probiotics. We found that the abundance profile of specific bacterial taxa and operational taxonomic groups (OTUs) varied across experimental groups.

## 2. Materials and Methods

### 2.1. Shrimp Aquaculture Production System

The research described in this report was conducted at the trū Shrimp Innovation Center (330 3rd Street, Balaton, MN, USA; 44.2° N 95.8° W), a research campus designed for the development of innovative indoor aquaculture techniques for shrimp production. Six polyethylene aquaculture production tanks (2.75 m × 6.7 m × 0.4 m) were used for this study, each controlled by an independent system to maintain temperature (28.0 ± 1.0 °C), provide water circulation (0.075−0.15 linear meters per second) and aeration (while rates varied according to production phase, dissolved oxygen levels were maintained at 4.5 mg/L or higher). 

Each tank contained approximately 2920 L of saltwater, with an average depth of 0.3 m. Salt water was prepared by dissolving Crystal Sea Marinemix (Marine Enterprises International, LLC., Baltimore, Maryland) with reverse osmosis-produced water at a final concentration of 28 g/L. Salinity was monitored and maintained at 28 ppt by adding approximately 58 L/day of reverse osmosis water to replace losses due to evaporation. Tank pH was kept at 7.7 ± 0.5 and alkalinity maintained at 150–300 ppm, with adjustments made by supplementation with sodium carbonate (9 mg /L of tank water for 0.5 unit reduction in pH below 7.4) or sodium bicarbonate (14 mg/L of tank water for 10 unit reduction in alkalinity), respectively. Fresh reverse osmosis water was added as needed to maintain tank depth.

Water from each tank was biologically filtered using separate floating bead bed bioreactors at a flow rate of 15 ± 0.5 L/m. Each bioreactor had a capacity of 0.015 m^3^ and consisted of a standard bio-medium of spherical beads, made of low-density polyethylene with a diameter of 1/8 inch and a surface area of 1100–1200 m^2^/m^3^ (catalog# BEADSFT3 Aquaculture Systems Technologies). Bioreactors were set up five days prior to the addition of shrimp in the production tanks by inoculation with 366 mL of a commercial bacterial stock (catalog# 75080590; FritzZyme Industries, Dallas, TX, USA), followed by daily supplementation with 8 g of ammonium chloride and 5 g of sodium nitrite to promote development of bacterial biofilms. 

Tanks were also inoculated with autotrophic bacteria five days prior to stocking using a commercial product consisting of *Nitrobacter* and *Nitrosococcus* species (0.135 mL stock per L of tank water; FritzZyme Industries, Dallas, TX, USA). Additional autotroph dosages were provided on trial days 1, 2, 12, 14, and 16. Further supplementations were provided when the total ammonia nitrogen (TAN) and/or nitrite levels exceeded concentrations of 5 mg/L and 2.5 mg/L, respectively.

Once bioreactor inoculation was completed, tanks were stocked with post-larval stage 12 shrimp, which were grown for 30 days. Following this period, all shrimp were removed from their respective tanks, pooled, then randomly redistributed at a stocking density of 2000 shrimp/tank (0.85 g ± 0.1 g). The stocking density (200 shrimp/m^3^, with a weight-based density of 0.175 ± 0.5 kg/m^3^) after redistribution was consistent with super-intensive production systems and was predicted to yield a final harvest density of 1.54 ± 0.5 kg/m^3^. Shrimp were then allowed to acclimate for 14 days before probiotic treatment (see below). The extruded feed (proprietary formulation) was manufactured by Prairie AquaTech (Brookings, SD, USA), with pellet diameters adjusted for shrimp growth, increasing from 1.8 to 2.4 mm during the course of the study. Feed ingredients included animal (fish and poultry meals), as well as plant products (soy and wheat meals); a proximate composition analysis of representative feed samples (Midwest Laboratories, Omaha, NE, USA) conducted prior to the experiment is presented in Table 1.

### 2.2. Experimental Design

The study consisted of two separate trials, with each trial conducted once using shrimp from a single genetic line. Two genetic lines of *Litopenaeus vannamei* were used: Shrimp Improvement Systems (SIS, Islamorada, FL, USA), selected for disease resistance, and Oceanic Institute (OI, Oahu, HI, USA), selected for growth. During each trial, three replicate tanks were supplemented with a commercial probiotic, while the remaining three tanks did not receive any supplementation (controls). Treatment tanks received a daily dose of the probiotic BioWish 3P (BioWish Technologies, Cincinnati, OH, USA) for 28 days, at a rate of 0.73 g/day. The probiotic was provided with the feed, which was offered over a 24 h period. The BioWish 3P product contained *Pediococcus acidilactici* (≥ 1 × 10^8^ cfu/g), *Pediococcus pentosaceus* (≥ 1 × 10^8^ cfu/g), *Lactobacillus plantarum* (≥ 1 × 10^8^ cfu/g), and *Bacillus subtilis* (≥ 1 × 10^7^ cfu/g). 

### 2.3. Analytical Methods

#### 2.3.1. Monitoring of Water Quality

Water chemistry testing was performed using industry validated methods (TAN method 8155 DR800, alkalinity method 10239 TNTplus, nitrite method 10019 DR800, LR, Test ‘N Tube, and nitrate method 8039 Cadmium Reduction). A Hach spectrophotometer (Hach Company, Loveland, CO, USA) was used for measurements. Total ammonia nitrogen (TAN), calculated un-ionized ammonia concentrations, as well as alkalinity were monitored daily. Nitrite and nitrate levels were measured three times every week (Mondays, Wednesdays, and Fridays). Dissolved oxygen, pH, temperature, and salinity were monitored daily using a YSI Professional Plus handheld multi-parameter meter (YSI Incorporated, Yellow Springs, OH, USA).

#### 2.3.2. Shrimp Performance and Health Assessments

Initial exams were performed from five randomly selected representative shrimp prior to stocking. Each selected individual was patted dry, weighted, then measured for total length (telson to the tip of the tail) and antenna length (head to the antenna tip). Body surface was then examined for necrosis or signs of injury, and gut fullness was assessed by examining the intestinal tract through the muscle wall, using a scale from 0 (100% full) to 4 (0–24% full). After pulling back the carapace, gills were then evaluated for presence of necrosis and/or debris using a scale from 0 (tissue completely clear of necrosis or debris) to 4 (15+ areas of necrosis and/or debris in a single viewing field). Finally, the hepatopancreas was dissected, weighed and examined for coloration and appearance, which were evaluated using a scale from 0 (no deformities, healthy organ) to 4 (16+ areas of severe tubular deformation, chronic ailment) [15].

During the trial, shrimp were sampled at seven-day intervals to monitor growth, performance and health. A total of 90 shrimp were netted, removed from the tank, then weighed individually. All but five randomly selected shrimp were returned to the production tank. Animal health exams as described above were then performed on the five selected shrimp.

The following health indices were calculated from the health exam data:

The mean condition factor (MCF), was determined with the equation [16]: (1)$$MCF=\frac{100*\left(Shrimp\;Weight\;\left(g\right)\right)}{{\left(Shrimp\;Length\;\left(cm\right)\right)}^{3}}$$

A MCF of less than 0.8 was indicative of low girth and poor health.

The hepatosomatic index (HSI) was calculated with the equation [17]: (2)$$HSI=\frac{Hepatopancreas\;Weight\;\left(g\right)}{Shrimp\;Weight\;\left(g\right)}$$

HSI is an assessment of a shrimp’s potential energy reserves; a value of 0.03 or lower indicated poor nutrient availability or absorption, while a value of 0.09 or above was a sign of possible systemic pathogenic infection.

The shrimp to antenna length ratio (SAR) was obtained with the following equation:(3)$$SAR=\frac{Average\;Antenna\;Length\;\left(cm\right)}{Shrimp\;Length\;\left(cm\right)}$$

A SAR score of 0.5 or lower was indicative of stress that could result in failure to thrive or death.

#### 2.3.3. Microbiological Analysis of Tank Water

Tank water samples were tested prior to stocking, then later on a weekly basis during the trial, to assess microbial populations for total heterotrophs, pathogenic *Vibrio*, as well as non-pathogenic *Vibrio*. Approximately 50 mL of tank water were collected with a sterile serological pipette and stored in a sterile 50 mL screw cap conical tube until analyzed by a commercial diagnostic laboratory (Research Technology Innovation Laboratory (RTI), Brookings, SD, USA). For diagnosis, serial dilutions were prepared to determine total heterotrophic counts by plating on marine-agar medium [18,19], as well as presumptive total *Vibrio* counts by plating on thiosulfate-citrate-bile salts-sucrose-agar medium. Potential pathogenic *Vibrio* colonies were distinguished from non-pathogenic by their color response on the selective media [20]. *Pediococcus* counts were determined by plating on De Man, Rogosa and Sharpe agar (MRS).

### 2.4. Microbial DNA Isolation and PCR Amplification

Shrimp gut samples were analyzed for bacterial composition on days 43 (prior to probiotic treatment, pre-treatment control), 57 (14 days probiotic treatment), and 71 (28 days probiotic treatment). Intestinal tissue was harvested from each animal using the following procedure. The telson was removed distal to the sixth abdominal segment with scissors, then the posterior end of the carapace was lifted to expose the hepatopancreas and the proximal end of the gut. The intestine was then excised with sterile tweezers starting at the hepatopancreas on through to the hind gut. Each sample consisted of intestinal tissue pooled from five individual shrimp from the same population to ensure sufficient material was available for DNA extraction. A total of 36 samples were harvested and stored with no preservative at −20 °C until DNA extraction.

Microbial DNA was isolated from gut samples using the repeated bead beating plus column method, as described by Yu and Morrison [21]. The V1-V3 region of the bacterial 16S rRNA gene was PCR-amplified using the 27F forward [22] and 519R reverse [23] primer pair. PCR reactions were performed with the Phusion Taq DNA polymerase (Thermo Scientific, Waltham, MA, USA) under the following conditions: hot start (4 min, 98 °C), followed by 35 cycles of denaturation (10 s, 98 °C), annealing (30 s, 50 °C) and extension (30 s, 72 °C), then ending with a final extension period (10 min, 72 °C). PCR products were separated by agarose gel electrophoresis, and amplicons of the expected size (~500 bp) were excised for gel purification using the QiaexII Gel extraction kit (Qiagen, Hilden, Germany). For each sample, approximately 400 ng of amplified DNA was submitted to Molecular Research DNA (MRDNA, Shallowater, TX, USA) for sequencing with the Illumina MiSeq 2X300 platform to generate overlapping paired-end reads.

### 2.5. Computational Analysis of PCR Generated 16S rRNA Amplicon Sequences

Unless specified, sequence data analysis was performed using custom written Perl scripts (available upon request). Raw bacterial 16S rRNA gene V1-V3 amplicon sequences were provided by Molecular Research DNA as assembled contigs from overlapping MiSeq 2 × 300 paired-end reads from the same flow cell clusters. Reads were then selected to meet the following criteria: presence of both intact 27F (forward) and 519R (reverse) primer nucleotide sequences, length between 400 and 580 nt, and a minimal quality threshold of no more than 1% of nucleotides with a Phred quality score lower than 15. 

Following quality screens, sequence reads were aligned, then clustered into operational taxonomic units (OTUs) at a genetic distance cutoff of 5% sequence dissimilarity [24]. While 3% is the most commonly used clustering cutoff for 16S rRNA, it was originally recommended for full length sequences, and may not be suitable for the analysis of specific sub-regions because nucleotide sequence variability is not constant across the entire length of the 16S rRNA gene. In this context, if 3% is a commonly accepted clustering cutoff for V4 or V4-V5 regions, which are the least variable of the hypervariable regions, then a higher cutoff should be used for the V1-V3 region, as V1 is the most variable region of the 16S rRNA gene. OTUs were screened for DNA sequence artifacts using the following methods. Chimeric sequences were first identified with the chimera.uchime and chimera.slayer commands from the MOTHUR open source software package [25]. Secondly, the integrity of the 5’ and 3’ ends of OTUs was evaluated using a database alignment search-based approach. When compared to their closest match of equal or longer sequence length from the NCBI nt database, as determined by BLAST [26], OTUs with more than 1% of nucleotides missing from the 5’ or 3’ end of their respective alignments were discarded as artifacts. Single read OTUs were subjected to an additional screen, where only sequences that had a perfect or near perfect match to a sequence in the NCBI nt database were kept for analysis, (i.e., the alignment had to span the entire sequence of the OTU, and a maximum of 1% of dissimilar nucleotides was tolerated). 

After removal of sequence chimeras and artifacts, taxonomic assignment of valid OTUs was determined using a combination of RDP Classifier [27] and BLAST [26]. The List of Prokaryotic Names with Standing in Nomenclature (LPSN—http://www.bacterio.net) was also consulted for information on valid species belonging to taxa of interest [28].

### 2.6. Computational Analysis for Microbial Community Diversity

For beta diversity analysis, abundance tables were first filtered by removing taxa found less than two times in 10% of the samples, and then a relative abundance table was made. Filtering allowed us to visualize high-level patterns in the dataset [29]. The data, based on Bray–Curtis distances [30], was ordinated by principal coordinates analysis (PCoA) [31]. The PCoA ordination matrix was generated using the ordinate function and plot by plot ordination function of the package “phyloseq” in R (R Foundation for Statistical Computing, Vienna, Austria). 

### 2.7. Statistical Analysis

Using R (Version R-3.2.3), the non-parametric Friedman test (command friedman.test) and the post hoc Nemenyi test for multiple pairwise comparisons (command posthoc.friedman.nemenyi.test) were performed to compare the abundance of bacterial taxonomic groups and OTUs between different groups of replicate samples, respectively. Means were considered to be significantly different when *p* ≤ 0.05.

### 2.8. Accession Numbers for Next Generation Sequencing Data

Raw sequence data are available from the NCBI Sequence Read Archive under Bioproject PRJNA551222.

## 3. Results

### 3.1. Comparative Analysis of Production Parameters and Growth Performance

No statistical differences (*p* > 0.05) were observed in performance (total feed offered, feed:gain, average daily gain or survival) between genetic lines or as a result of probiotic treatment. Similarly, no differences in health indices, i.e., mean condition factor (MCF), shrimp-antenna length ratios (SAR) or hepatosomatic index (HSI), were found across samples (Appendix A).

Water chemistry was monitored and managed throughout the trial. Overall, total ammonia nitrogen (TAN), unionized ammonia (UA), and nitrite were numerically higher in OI production tanks compared to SIS production tanks, but these differences were not found to be statistically significant (*p* > 0.05; Supplementary Tables S1 and S2). Similarly, other measured water chemistry parameters (nitrate, pH, and alkalinity; Appendix A) did not vary significantly during the trial. Microbial quality of production tank water was assessed using culture-based microbiological assays. No significant differences were found for total heterotrophic counts (*p* > 0.05). While pathogenic *Vibrio* were not detected using this method, non-pathogenic *Vibrio* species were found at an average density ranging between 4.10 × 10^3^ and 5.47 × 10^3^ CFU/ml (*p* > 0.05), which is within a range consistent with normal operating conditions for indoor shrimp aquaculture production. Counts for *Pediococcus* ranged between 2.41 × 10^2^ and 8.25 × 10^2^ CFU/ml across all tanks, with no statistical supported differences between genetic lines or as a result of probiotic treatment (*p* > 0.05, Appendix A).

### 3.2. Comparative Analysis of Gut Bacterial Communities by Taxonomic Composition

To investigate the potential effect of host genetic background and/or probiotic treatment on the composition of gut bacterial communities, an analysis using the 16S rRNA gene as a phylogenetic marker was performed. A total of 609,210 high quality and chimera/artifact-free reads were generated across 36 samples, with an average of 13,569 ± 8158 reads per sample for the SIS genetic line and 20,276 ± 11,064 reads per sample for the OI genetic line. 

Consistent with previous published reports on bacterial communities of the shrimp gut, Proteobacteria was found to be the most abundant phylum in this study, with means of experimental groups ranging between 43.68 and 80.84% (Table 2). Within Proteobacteria, Vibrionales and Rhodobacterales were the most highly represented orders, together accounting for 92.78 to 99.44% of Proteobacteria in individual samples. While found in lower abundance, the phyla Bacteroidetes (1.14–45.98%), Firmicutes (0.42–50.13%) and Verrucomicrobia (0.51–28.80%) were overall well represented across experimental groups. Other minor phyla, such as Planctomycetes (0.34–2.97%) and Saccharibacteria (0.02–4.87%), were also identified.

Among the eight taxonomic groups described above, six were found to vary significantly across experimental groups (*p* < 0.05). While they were not supported by the Nemenyi multiple comparison test, a number of taxonomic groups exhibited abundance patterns that were suggestive of a response to treatment or trial parameters. Notably, Rhodobacterales were found at their highest abundance in the SIS line on day 43, i.e., after completion of the adaptation period and prior to the addition of the probiotic, then decreased during the following 28 days. In contrast, levels of Rhodobacterales did not exhibit the same degree of variation in the OI line. Verrucomicrobia exhibited an opposite profile, with highest levels in the OI genetic line on day 43, followed by lower abundance at later time points. In contrast, the abundance of the members of this phylum was not found to vary to the same extent across samples for the SIS genetic line. In the OI genetic line, members of the phylum Firmicutes were found in higher abundance in samples collected on days 57 and 71 compared to day 43, with means of 22.10–37.07% on day 71. Firmicutes in SIS shrimp guts were in contrast present in very low abundance, with means ranging from 0.42 to 0.83. Bacteroidetes in the SIS genetic line were found to increase after the day 43 time point, with abundances in the absence of probiotics observed to be 3.5–3.8 times higher than in probiotic-treated shrimp at the same time points.

### 3.3. Comparative Analysis of Gut Bacterial Communities by OTU Composition

A total of 2195 OTUs were identified across all samples (Appendix A), with 707 corresponding to previously described OTUs [32]. OTUs that were common to both SIS and OI genetic lines represented the vast majority (99.3–99.7%) of sequence reads generated in the present study (Figure 1). However, because the taxonomic analysis, as described in Section 3.2, had indicated potential differences in composition between genetic lines, these combined results suggested at that point that SIS and OI shrimp shared common OTUs that were present at different abundance levels in their respective genetic lines. To gain further insight, principal coordinate analysis (PCoA) was performed to investigate beta diversity. From this analysis, samples were found to cluster in a pattern that was indicative of differences in OTU composition between genetic lines (Figure 2A). PCoA results also suggested that changes in OTU composition had occurred in both genetic lines between the beginning and the end of the trial (Figure 2B). 

To further investigate the changes in bacterial community composition that had occurred during the trial, the distribution profiles of the most abundant OTUs were analyzed (Table 3). Five of these major OTUs (SD_Shr-00002, SD_Shr-00003, SD_Shr-00004, SD_Shr-00006 and SD_Shr-00010) had previously been described by our group [32], while SD_Shr-00097 (Bacteroidetes-affiliated) and SD_Shr-00098 (Verrucomicrobia-affiliated) were novel OTUs. While SD_Shr-00098 was very closely related to *Haloferula rosea* (99.4%), the sequence identity of SD_Shr-00097 to its closest valid relative was only 93.2% (*Salinimicrobium catena*) indicating that it likely belonged to a bacterial phylogenic lineage that has yet to be described. SD_Shr-00097 was found to be almost identical (99.4%) to the 16S rRNA sequence of an uncultured bacterial species that was identified in a recirculating mariculture system (GenBank sequence JX306764). The abundance of five of the major OTUs was found to vary significantly across experimental groups (*p* < 0.05), with limited pairwise differences in abundance supported by the Nemenyi multiple comparison test. For Proteobacteria-affiliated main OTUs, two distinct patterns were observed. SD_Shr-00006 was in highest abundance in the SIS samples collected on d43, which was greater by 7.3–7.9-fold than d57 samples and by 21.9–24.3-fold than d71 samples of the same genetic line. While a decrease with time was observed, the profile for this OTU was not as pronounced in the OI genetic line. SD_Shr-00004 showed an opposite profile to SD_Shr-00006, with highest abundance in the SIS d71 samples and lowest in the SIS d43 samples, either with probiotic supplementation (10.3 times higher abundance) or without (13.9 times higher abundance). SD_Shr-00097 displayed an intriguing abundance profile, with highest abundance in samples from SIS production tanks at day 57 and day 71 that were not supplemented with probiotics. This SD_Shr-00097 profile was not observed in the OI genetic line. For the Verrucomicrobia affiliated OTU SD_Shr-00098, the highest abundance was observed in the day 43 samples from the OI genetic line, with much lower levels in day 71 samples, by a factor of 35.1 to 49.5-fold. This pattern was not observed in SIS shrimp.

## 4. Discussion

In the shrimp industry, there has been an increasing reliance on aquaculture to provide a steady and safe source of product. While this represents an attractive opportunity for expansion, it has also raised concerns and challenges with regards to animal health, mitigation of disease outbreaks, as well as environmental and economic sustainability. Compared to pond systems, indoor facilities have the advantage of providing better control of ambient conditions and reduced exposure to pathogens. However, because they require more costly infrastructure to operate, indoor production systems need to pay close attention to other management practices in order to remain economically competitive. A common strategy has been to use feed formulations with higher inclusion of plant-based products, as they are less costly than traditionally used ingredients such as fish meals and oils. However, plant-based products include a higher concentration of polysaccharides than what is typically found in diets of shrimp living in natural environments. Because most shrimp health problems are nutritionally related [9], feeding plant ingredients could affect animal health, thus potentially reducing disease resistance in indoor shrimp. This could pose a serious problem, particularly if genetic lines that favor growth over disease resistance are used in production. In addition, minimizing exposure with natural environments may prevent gut colonization with critical symbionts that would provide long term health and/or nutritional benefits to indoor-grown shrimp, as has been described in humans and other animals.

Considering the reported importance of the gut microbiome in animal health and nutrition, the shrimp aquaculture industry would greatly benefit from developing strategies to modulate the composition and metabolic activities of gut symbionts. However, the development of effective approaches or products requires improved insight and knowledge of the shrimp microbiome. To this end, the study presented in this report aimed at investigating the potential effects of the genetic background and probiotic treatment on the composition of the developing gut bacterial communities in shrimp raised under standard indoor operating management procedures, with all production tanks provided with the same feed formulation. While there were no statistically supported differences in performance or animal health associated with genetics or probiotic treatment under the production conditions used in this study, a number of effects on the composition of gut bacterial communities in indoor grown shrimp were observed. Notably, major differences in bacterial composition were observed between the two genetic lines. For instance, while SD_Shr-00006 (Proteobacteria) and SD_Shr-00098 (Verrucomicrobia) showed a similar composition pattern, with higher abundance at day 43 compared to day 71 samples, their respective profiles were observed in different genetic lines. Similarly, higher abundance of SD_00097 in shrimp from tanks not supplemented with probiotics was observed in the SIS lines but not in the OI genetic line. 

The contrast between genetic lines for the abundance of SD_Shr-00006 and SD_Shr-00098 at the day 43 time point was quite striking, considering that all shrimp had been reared under the same conditions and provided with the same feed. It is thought that the initial development of the gut microbiome for shrimp is primarily acquired from artemia, a live feed that is offered shortly after hatching [9]. Bacterial phyla that dominate the shrimp gut at this stage include Proteobacteria, Bacteroidetes and Actinobacteria [9]. The microbiome then continues to develop as its host grows, and its composition is influenced by a number of factors that include dietary ingredients and developmental stage of the host. While other parameters, such as shrimp health status as well as environmental conditions (e.g., water chemistry and water microbiology), can also affect the composition of gut microbial communities [12], they would not be expected to have played an impactful role in this study, as no statistically supported differences for these conditions were observed. 

Another surprising observation from this study was the effect of probiotic supplementation on the composition of shrimp gut bacterial communities. Indeed, SD_Shr-00097 was found in highest abundance in the absence of probiotic treatment (SIS line), while SD_Shr-00010 showed its highest abundance in samples from probiotic supplemented tanks. Probiotic supplementation is a common practice in indoor shrimp aquaculture as a source of microorganisms, i.e., as a means to mimic natural aquatic systems that harbor beneficial bacteria, such as the genera *Lactobacillus, Streptococcus*, and *Bacillus*, in sediments and water [10]. Some studies have reported that stress can reduce growth of these bacteria and other natural microbiota, allowing uncontrolled growth of pathogenic bacteria and eventually disease [12]. As *Pediococcus*, *Lactobacillus*, and *Bacillus*-affiliated OTUs were found in only very low abundance or were undetectable in the gut of probiotic supplemented shrimp, bacterial species from the commercial probiotic formulation did not appear to efficiently colonize the shrimp gut in this study. However, their presence or absence did impact the development of gut bacterial communities. While future investigations will be necessary to uncover the mechanisms involved, it could be hypothesized that, even if probiotic bacterial species do not become established in high density in the gut of exposed shrimp, they produce metabolites that favor the establishment of certain bacterial species or OTUs over others.

Opportunistic pathogens tend to always be present in the gut of healthy animals in low abundance. They remain benign under normal conditions [13], with disease occurring when conditions such as accumulating stress or poor diets favor their proliferation. In shrimp, the most common pathogens belong to the family Enterobacteriaceae; these include species of the genera *Pseudomonas*, *Flavobacterium*, *Escherichia*, *Aeromonas*, *Vibrio*, *Rickettsia*, *Shewanella*, and *Desulfovibrio* [9]. Of these genera, only *Vibrio* was identified in this study. Intriguingly, not only were *Vibrio*-related sequence reads very abundant across all experimental groups, their two most abundant OTUs were most closely related to *Vibrio alginolyticus* (SD_Shr-00004) and *Vibrio shilonii* (SD_Shr-00010). Both species are known to cause disease in shrimp, with *V. alginolyticus* thought to be more pathogenic by resulting in failure to thrive and ultimately death of infected hosts, while *V. shilonii* infection is more likely to cause tissue damage and reduced weight gain [33]. Interestingly, SD_Shr-00004 had previously been identified by our group as prominent in outdoor systems while found in very low abundance in indoor-aquaculture raised shrimp, which had suggested that husbandry practices may play an important role in controlling the abundance of this OTU [32]. However, SD_Shr-00004 was found in much higher abundance in this study. Intriguingly, the highest levels were observed in the SIS genetic line, which has been bred for disease resistance, and not the OI genetic line, which was bred for growth. Similarly, SD_Shr-00010 was found in high abundance in both genetic lines, with no statistically supported differences between them. These observations suggest that these OTUs may not correspond to a pathogenic strain of their respective closest relatives. 

Together, the results presented in this report suggest that host genetic background and the use of probiotics can affect the gut bacterial composition of aquaculture-raised whiteleg shrimp. They further indicate that development of strategies to effectively manipulate the microbiome of this important seafood will need to take into consideration the genetic background of the various lines used in aquaculture production.

## Figures and Tables

**Figure 1 microorganisms-07-00217-f001:**
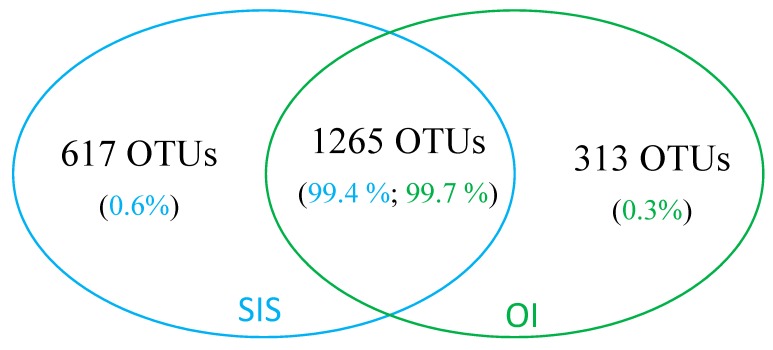
Venn diagram showing the number of shared and unique intestinal bacterial OTUs between the SIS and OI genetic lines of whiteleg shrimp raised in an indoor facility. Also shown is the proportion of sequence reads for each category.

**Figure 2 microorganisms-07-00217-f002:**
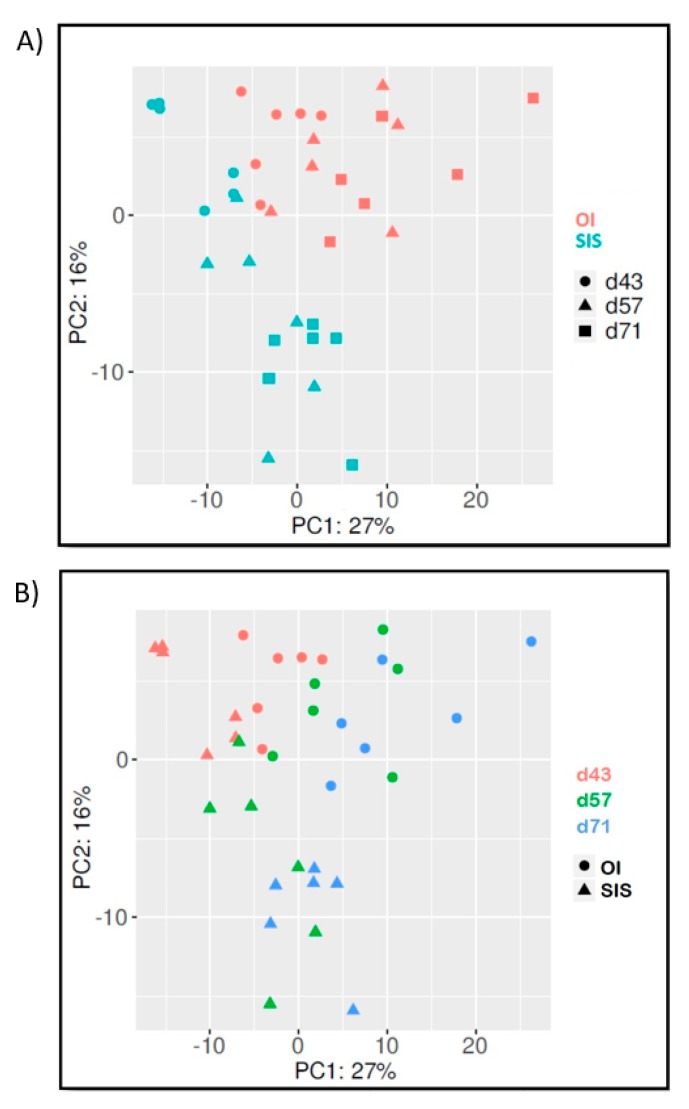
Comparison of intestinal bacterial communities from whiteleg shrimp using principle coordinate analysis (PCoA). The x and y axes correspond to principal components 1 (PC1) and 2 (PC2), which explained the highest level of variation. Both panels show the same ordination graph, highlighting differences in profile either between genetic lines (**A**) or tenure in production tank (**B**).

**Table 1 microorganisms-07-00217-t001:** Proximate analysis for proprietary diet fed to all research tanks.

Analyzed Nutrients	Units	Observed Value
Moisture	%	6.85
Dry Matter	%	93.15
Protein (crude)	%	37.50
Fat (crude)	%	9.54
Fiber (crude)	%	1.80
Ash	%	11.40
Digestible Energy	Mcal/lbs	1.55
Total Digestible Nutrients	%	77.10
Metabolizable Energy	Mcal/lbs	1.36
Net Energy (gain)	Mcal/lbs	0.56
Net Energy (lactation)	Mcal/lbs	0.81
Net Energy (maint.)	Mcal/lbs	0.84

**Table 2 microorganisms-07-00217-t002:** Mean relative abundance (%) of main bacterial taxonomic groups in the intestinal tract of whiteleg shrimp from two genetic lines (SIS or OI), in the presence (+) or absence (−) of probiotic treatment, at three different sampling time points (d43, d57 and d71).

Taxonomic Group	SIS.43	SIS.57+	SIS.57−	SIS.71+	SIS.71−	OI.43	OI.57+	OI.57−	OI.71+	OI.71−	*P* Values *
**Proteobacteria**	77.21	80.84	43.68	87.22	64.93	62.96	79.70	47.35	73.61	56.43	0.09470
Rhodobacterales	47.41^bc^	8.79 ^a^	11.13 ^a^	3.96 ^a^	9.63 ^a^	12.94 ^a^	18.83 ^ac^	1.86 ^a^	4.43 ^a^	6.21 ^a^	0.00032
Vibrionales	27.93	71.60	31.95	82.70	54.44	47.76	59.04	45.01	63.86	48.19	0.07790
Other Proteobacteria	1.86	0.46	0.60	0.57	0.86	2.26	1.83	0.48	5.32	2.03	ND ^#^
**Bacteroidetes**	4.80 ^ac^	11.99 ^ac^	45.98 ^b^	8.59 ^ac^	30.57 ^bc^	4.05 ^a^	5.24 ^ac^	1.14 ^ac^	1.47 ^ac^	4.49 ^ac^	0.00018
**Verrucomicrobia**	7.28 ^a^	3.66 ^a^	8.22 ^ac^	2.48 ^a^	2.53 ^a^	28.80 ^bc^	4.87 ^ac^	0.51 ^a^	1.01 ^a^	0.80 ^a^	0.00348
**Firmicutes**	0.47 ^a^	0.70 ^a^	0.42 ^a^	0.83 ^a^	0.48 ^a^	1.65 ^a^	7.56 ^ab^	50.13 ^b^	22.10 ^ab^	37.07 ^ab^	0.00159
**Planctomycetes**	2.97 ^b^	0.57 ^ab^	0.82 ^ab^	0.42 ^a^	0.80 ^ab^	2.09 ^ab^	0.51 ^a^	0.34 ^a^	0.87 ^ab^	0.50 ^a^	0.00435
**Saccharibacteria**	4.87	0.41	0.13	0.07	0.22	0.06	1.02	0.02	0.25	0.09	0.13700
**Other Phyla**	2.17	0.42	0.54	0.28	0.39	0.18	1.02	0.49	0.56	0.39	ND ^#^

^a, b, c^ Values statistically different from each other based on the post hoc Nemenyi test are distinguished by different superscripts; * determined by the Friedman test; ^#^ the Friedman test was not performed for these groups because they included multiple ranks of the same taxonomic level (i.e., orders or phyla).

**Table 3 microorganisms-07-00217-t003:** Mean relative abundance (%) of main bacterial operational taxonomic units in the intestinal tract of whiteleg shrimp from two genetic lines (SIS or OI), in the presence (+) or absence (−) of probiotic treatment, at three different sampling time points (day 43, day 57 and day 71).

OTUs	SIS.43	SIS.57+	SIS.57−	SIS.71+	SIS.71−	OI.43	OI.57+	OI.57−	OI.71+	OI.71−	*P* Values *
**Proteobacteria**											
SD_Shr-00002	6.89 ^bc^	2.24 ^a^	3.07 ^ac^	0.83 ^a^	1.91 ^a^	1.73 ^a^	2.48 ^a^	0.56 ^a^	0.59 ^a^	0.68 ^a^	0.00011
SD_Shr-00004	2.38 ^a^	7.45 ^ac^	7.22 ^ac^	24.58 ^bc^	33.16 ^b^	2.09 ^a^	3.88 ^a^	3.03 ^a^	6.03 ^a^	5.63 ^a^	6.22 × 10^−6^
SD_Shr-00006	21.91 ^b^	2.78 ^a^	3.02 ^a^	1.00 ^a^	0.90 ^a^	6.82 ^a^	4.54 ^a^	0.39 ^a^	1.08 ^a^	2.92 ^a^	0.000824
SD_Shr-00010	22.02	58.69	21.32	50.01	14.78	40.06	49.16	38.13	48.11	34.66	0.114
**Firmicutes**											
SD_Shr-00003	0.39 ^a^	0.52 ^a^	0.33 ^a^	0.64 ^a^	0.38 ^a^	1.56 ^a^	2.81 ^a^	49.00 ^b^	21.44 ^ab^	36.14 ^ab^	0.00161
**Bacteroidetes**											
SD_Shr-00097	1.95 ^a^	8.13 ^a^	40.11 ^bc^	2.11 ^a^	25.68 ^ac^	3.13 ^a^	2.46 ^a^	0.68 ^a^	0.60 ^a^	2.30 ^a^	0.000345
**Verrucomicrobia**											
SD_Shr-00098	2.50 ^a^	1.26 ^a^	6.43 ^ab^	0.42 ^a^	0.83 ^a^	25.27 ^b^	4.18 ^ab^	0.41 ^a^	0.72 ^a^	0.51 ^a^	0.00626

^a, b, c^ Values statistically different from each other based on the post hoc Nemenyi test are distinguished by different superscripts; * determined by the Friedman test; ^#^ the Friedman test was not performed for these groups because they included multiple ranks of the same taxonomic level (i.e., orders or phyla).

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
