# Peer review of "Investigation of the Potential Effects of Host Genetics and Probiotic Treatment on the Gut Bacterial Community Composition of Aquaculture-raised Pacific Whiteleg Shrimp, Litopenaeus vannamei"

_microorganisms, 2019, doi:10.3390/microorganisms7080217_

Reviewer 1 Report

Landsman et al. presented a report on testing the effect of probiotics on changing gut microbiota of commercially raised Pacific whiteleg shrimp. The concept is interesting to the industry, and the paper was well written. However, there are several issues that need to be addressed before this manuscript can be published.

Major issues:

1. The authors conducted separate trials for SIS and OI, and therefore these two lineages cannot be compared, as the difference could be be explained by different experimental conditions and other technical/environmental differences between two separate trials. If the authors intended to claim the genetic difference between two lineages, the two trials needed to be conducted as one trial including two lineages. It's obviously much difficult to do, in terms of logistics and cost, but the distinction needs to be made clear. Along similar lines, the title of this manuscript is not appropriate, because this study actually tested the effect of probiotics on changing gut microbiota, not host genetics. 

2. Because of the experimental design, regular ANOVA should not be used, and the repeated measures ANOVA should be used instead, to account for the non-independence between sampling points. The assumptions of repeated measures ANOVA should be tested before performing the test, namely normality and sphericity. If these assumptions cannot be met, data transformation and/or non-parametric approaches need to be considered. 

3. Because of the compositional constraints of adding up to 1, the authors might need to take that into account when comparing relative abundances. Methods like ANCOM (analysis of composition of microbiomes) and ALDEx (ANOVA-like differential express) might help in this case.

Minor issues: 

1. The gut fullness scale is confusing and counter-intuitive, as 100% full is zero, and 1-20% full is 4. 

2. Why was this particular probiotic chosen for this study, would other probiotics elicit similar changes? Are the changes elicited by the probiotic biologically relevant?

3. Figure 2 A and B show the same plot with different representations, which is not necessary and redundant. 

4. Line 347, one word missing

Author Response

Manuscript: microorganisms-549479

Original Title: Host genetics is an important determinant of gut bacterial community composition in aquaculture-raised Pacific Whiteleg shrimp, Litopenaeus vannamei

Revised Title: Investigation of the potential effects of host genetics and probiotic treatment on the gut bacterial community composition of aquaculture-raised Pacific Whiteleg shrimp Litopenaeus vannamei

Authors: Angela Landsman, Benoit St-Pierre, Misael Rosales-Leija, Michael Brown, William Gibbons

Rebuttal to reviewer comments

Reviewer 1

Landsman et al. presented a report on testing the effect of probiotics on changing gut microbiota of commercially raised Pacific whiteleg shrimp. The concept is interesting to the industry, and the paper was well written. However, there are several issues that need to be addressed before this manuscript can be published.

We thank you for the positive feedback. We hope that we have addressed your concerns and revised the manuscript to your satisfaction.

Rev1-Comment1

The authors conducted separate trials for SIS and OI, and therefore these two lineages cannot be compared, as the difference could be explained by different experimental conditions and other technical/environmental differences between two separate trials. If the authors intended to claim the genetic difference between two lineages, the two trials needed to be conducted as one trial including two lineages. It's obviously much difficult to do, in terms of logistics and cost, but the distinction needs to be made clear. Along similar lines, the title of this manuscript is not appropriate, because this study actually tested the effect of probiotics on changing gut microbiota, not host genetics.

Agreed. We have revised the title of the manuscript, which now reads:

“Investigation of the potential effects of host genetics and probiotic treatment on the gut bacterial community composition of aquaculture-raised Pacific Whiteleg shrimp Litopenaeus vannamei “

Certain passages elsewhere in the revised manuscript were also edited to address this concern (highlighted and labeled as addressing this comment).

Rev1-Comment2

Because of the experimental design, regular ANOVA should not be used, and the repeated measures ANOVA should be used instead, to account for the non-independence between sampling points. The assumptions of repeated measures ANOVA should be tested before performing the test, namely normality and sphericity. If these assumptions cannot be met, data transformation and/or non-parametric approaches need to be considered.

Agreed. We have reanalyzed the data using the Friedman test (non parametric) with a Nemenyi post hoc test for multiple pairwise comparisons. Compared to the original analysis, the Friedman test revealed that Saccharibacteria varied significantly, while OTU SD-Shr_00097 was not found to vary significantly; the respective status (i.e. significantly different or not) of the remaining taxa and OTUs as described in the original manuscript was not changed. As a result of the post hoc Nemenyi test, however, only a limited number of pairwise comparisons were found to be statistically different.

Changes reflecting the reanalysis (highlighted and labeled as addressing this comment) were made as necessary throughout the revised manuscript.

Rev1-Comment3

Because of the compositional constraints of adding up to 1, the authors might need to take that into account when comparing relative abundances. Methods like ANCOM (analysis of composition of microbiomes) and ALDEx (ANOVA-like differential express) might help in this case.

It is our understanding that the main concern associated with “compositional constraints of adding up to 1” is that removal of sequences from a given dataset would a)  change the relative abundance of the remaining groups as part of the dataset to analyze and b) consequently change the ratio of abundances between these groups.

In our approach, removal of sequences from datasets to analyze (i.e. when the effect described above would be most likely to take place) happens during 2 main steps: quality filtering and screening for chimera and sequence artifacts. For a given dataset, we have observed that the percentage of reads that are removed as a result of either step are within the same range across samples, so we would not expect dramatic shifts in abundance of specific groups as a result of these steps. Furthermore, we have found no indication that there is removal of specific groups of closely related sequences (presumptive OTUs) as a result of these steps.

We would consider this a major concern if the most abundant OTUs were removed from the analysis, as their removal would be more likely to cause major shifts in abundance profiles. As far as we can tell, there is no indication that the sequences that are removed are from a particular group or cluster. We are careful to monitor the number of sequences that remain after each step of our pipeline to determine if unusually high numbers of sequences are removed at any given point of the analysis.

In the paper by Fernandes et al (2014, Microbiome, 2, 15) describing ALDEx, a theoretical example of this type of effect is presented, where a group of 500 sequences from a total of 595 sequences is removed from the analysis. The only situation where we envision that this could happen is if the clustering approach used was based on reference OTUs; in this situation, large groups of closely related sequences that belonged to OTUs not present in the reference list could be discarded. However, a de novo OTU clustering approach, as used in our pipeline, would avoid this type of problem.  

Rev1-Comment4

The gut fullness scale is confusing and counter-intuitive, as 100% full is zero, and 1-20% full is 4.

While we understand the reviewer’s point of view, the standards were referenced directly from Dr. Lightner’s former practices as one of the world’s leading shrimp pathologist.

Rev1-Comment5

Why was this particular probiotic chosen for this study, would other probiotics elicit similar changes? Are the changes elicited by the probiotic biologically relevant?

The probiotic utilized was recommended by researchers from Texas A&M University; their anticipated effect was to provide protection (by competitive exclusion) against pathogens that can be found ubiquitously in the water column.  

As expressed by the reviewer, it remains to be determined whether the changes observed in response to probiotic treatment are ‘biologically relevant’. We hope that the research presented in this report will help in guiding future efforts to address this question.

Rev1-Comment6

Figure 2 A and B show the same plot with different representations, which is not necessary and redundant.

We presented two versions of the same plot to facilitate visualizations of two different profiles.  As we prefer to keep both versions as they were in the original manuscript, we propose to leave this decision to the editor’s discretion.

Rev1-Comment7

 Line 347, one word missing

Thank you for noting this omission, which seems to have occurred during formatting. The word “beta” has been inserted in the revised manuscript (Revised manuscript, line 350).

Reviewer 2 Report

The content of the submitted manuscript is scientifically sound, well presented and discussed. The work describes a complex and detailed study which focuses on the effects of the genetic line and the use of probiotics on the gut microbiome of Pacific whiteleg shrimp reared in indoor aquaculture facility. The study shows a lot of observations and in some cases it needs some clarifications:

Abstract

Lines 32: “Notably, SD­­_Shr-00006 was……..” (For starting to present the results it would be better to specify OTUs) .

Introduction

Lines 112: “We found……between genetic lines” (I would put in the results).

2 Materials and Methods

2.3.3.  Microbiological Analysis

Lines 213-218: “For diagnosis………….to determine……..total Vibrio counts on TCBS………………..”. You can count only presumptive Vibrio by TCBS”.

2.4. Microbial DNA Isolation and PCR Amplification

Lines 227-228: How many samples in total? You say 36 in the results but you should indicate it also here.

Results

Lines 336-338: Bacteroidetes in the SIS genetic line………….., with abundances in probiotici treated samples found to be 3.5-3.8 X higher? than the no probiotic controls……. From table 2 mean abundances in probiotic treated samples are 3.5-3.8 lower than the no probiotic controls.

Lines 343: However………, as described in section 4.2? It is section 3.2.

Discussion

Lines 420: “notably, all differences in bacteria composition were…….between the two genetic lines”. The term all is excessive in fact the gut microbiota is a very complex ecosystem which depends on various factors including, first of all, food (as you stated), the environment (water, temperature etc.) and  host genetics. In various cases the differences of the gut microbiome in the genetic lines are not significant.

Lines 424: “similarly, higher abundance of SD 00097in shrimp from probiotic supplemented tanks?......”This is true for tanks not inoculated with probiotics”.

After a minor revision I recommend to the Editor the publication of this work.

Author Response

Manuscript: microorganisms-549479

Original Title: Host genetics is an important determinant of gut bacterial community composition in aquaculture-raised Pacific Whiteleg shrimp, Litopenaeus vannamei

Revised Title: Investigation of the potential effects of host genetics and probiotic treatment on the gut bacterial community composition of aquaculture-raised Pacific Whiteleg shrimp Litopenaeus vannamei

Authors: Angela Landsman, Benoit St-Pierre, Misael Rosales-Leija, Michael Brown, William Gibbons

Rebuttal to reviewer comments

Reviewer 2

The content of the submitted manuscript is scientifically sound, well presented and discussed. The work describes a complex and detailed study which focuses on the effects of the genetic line and the use of probiotics on the gut microbiome of Pacific whiteleg shrimp reared in indoor aquaculture facility. The study shows a lot of observations and in some cases it needs some clarifications.

Thank you, we very much appreciate your positive feedback.

Rev2-Comment1

Abstract

Lines 32: “Notably, SD­­_Shr-00006 was……..” (For starting to present the results it would be better to specify OTUs) .

Done. We have added the full definition in the abstract. The revised MS now reads:

Line 33: “Notably, operational taxonomic unit (OTU) SD_Shr-00006 was (…)”

Rev2-Comment2

Introduction

Lines 112: “We found……between genetic lines” (I would put in the results).

While we acknowledge that traditional scientific writing discourages including results from an article in the Introduction, we like to present a very brief summary of the findings as a lead in to the results for the benefit of the reader.

Rev2-Comment3

2 Materials and Methods

2.3.3.  Microbiological Analysis

Lines 213-218: “For diagnosis………….to determine……..total Vibrio counts on TCBS………………..”. You can count only presumptive Vibrio by TCBS”.

Thank you for recommending this clarification. We have added the word presumptive in section 2.3.3 (Materials and Methods). The revised MS now reads:

Lines 214-217: “For diagnosis, serial dilutions were prepared to determine total heterotrophic counts by plating on marine-agar medium, as well as presumptive total Vibrio counts by plating on thiosulfate-citrate-bile salts-sucrose-agar medium.”

Rev2-Comment4

2.4. Microbial DNA Isolation and PCR Amplification

Lines 227-228: How many samples in total? You say 36 in the results but you should indicate it also here.

Agreed. We have added sample count in section 2.4 (Materials and Methods). The revised MS now reads:

Lines 228: “A total of 36 samples were harvested and stored with no preservative at -20°C until DNA extraction.”

Rev2-Comment5

Results

Lines 336-338: Bacteroidetes in the SIS genetic line………….., with abundances in probiotic treated samples found to be 3.5-3.8 X higher? than the no probiotic controls……. From table 2 mean abundances in probiotic treated samples are 3.5-3.8 lower than the no probiotic controls.

Thank you for noticing this error. We have corrected the statement, and the revised MS now reads:

Lines 339-341: “Bacteroidetes in the SIS genetic line were found to increase after the day 43 time point, with abundances in the absence of probiotics observed to be 3.5 -3.8 X higher than in probiotics-treated shrimp of the same time points.”

Rev2-Comment6

Lines 343: However………, as described in section 4.2? It is section 3.2.

Thank you for bringing this to our attention. We have corrected 4.2 to 3.2 in the Results. The revised MS now reads:

Line 346: “However, …, as described in section 3.2, had indicated differences in composition (…)”

Rev2-Comment7

Discussion

Lines 420: “notably, all differences in bacteria composition were…….between the two genetic lines”. The term all is excessive in fact the gut microbiota is a very complex ecosystem which depends on various factors including, first of all, food (as you stated), the environment (water, temperature etc.) and host genetics. In various cases the differences of the gut microbiome in the genetic lines are not significant.

Agreed. We have revised the statement, which now reads:

Line 423: “Notably, major differences in bacterial composition were observed between the two genetic lines.”

Rev2-Comment8

Lines 424: “similarly, higher abundance of SD 00097in shrimp from probiotic supplemented tanks?......”This is true for tanks not inoculated with probiotics”.

Agreed. The revised MS now reads:

Lines 427-428: “(…) higher abundance of SD_00097 in shrimp from tanks not supplemented with probiotics was observed in the SIS lines but not in the OI genetic line.”

After a minor revision I recommend to the Editor the publication of this work.

We very much appreciate your support.  Thank you for your time and comments towards improving our work.